# Measuring the Effect of an Ergonomic Lecture on the Rapid Upper Limb Assessment Scores of Dental Assistant Students Using Inertial Sensor-Based Motion Capture—A Randomized Controlled Study

**DOI:** 10.3390/healthcare12161670

**Published:** 2024-08-21

**Authors:** Steven Simon, Laura Laurendi, Jonna Meining, Jonas Dully, Carlo Dindorf, Lukas Maurer, Michael Fröhlich

**Affiliations:** 1Department of Sports Science, University of Kaiserslautern-Landau, 67663 Kaiserslautern, Germany; jonas.dully@rptu.de (J.D.); carlo.dindorf@rptu.de (C.D.); l_maurer@rptu.de (L.M.); michael.froehlich@rptu.de (M.F.); 2Medical Department, Dental Practice Dr. Laurendi, 67547 Worms, Germany; laulaurendi@gmail.com; 3Department of Education, Worms BBS Dental School, 67547 Worms, Germany; jomeining@aol.de

**Keywords:** ergonomics, motion capture, Nordic Questionnaire, work-related discomfort, dental assistant health, observational methods, artificial intelligence

## Abstract

Individuals working in the field of dentistry have a high prevalence of musculoskeletal disorders (MSDs) owing to monotonous and one-sided physical exertion. Inertial measurement units (IMU) are increasingly shifting into focus for assessing postural risk at work. Therefore, the present study aimed to evaluate the effects of an ergonomic lecture and training intervention on postural risk and MSDs in dental assistant students using inertial sensor-based motion capture (MoCap). Eighteen female dental assistant students (age: 19.44 ± 6.83 years; height: 164.59 ± 5.32 cm; weight: 64.88 ± 16.52 kg; BMI: 19.70 ± 4.89 kg/m^2^), randomly divided into intervention (n = 9) and control (n = 9) groups, participated in the present study. The participants completed the Nordic Questionnaire on MSD prevalence, after which a 90 s MoCap with Xsens IMU was performed. A lecture on ergonomics was provided, followed by a five-week intervention for the intervention group. Follow-up assessments were performed, and 5- and 18-week follow-up MSD questionnaires were administered. Mixed analysis of variance (MANOVA) showed a significant difference in the Rapid Upper Limb Assessment (RULA) and part-scores of the upper arm and wrist. Despite a reduction in MSDs, no significant differences in the time of measurement and groups were detected after the five-week training intervention and the 18-week follow-up questionnaire. A targeted ergonomics lecture was effective for dental assistant students, and technologies such as IMU improved workplace ergonomics in dentists. Further studies with a longer measurement periods, follow-up, and larger sample sizes are recommended.

## 1. Introduction

Day-to-day work in the field of dentistry requires physical exertion, making the musculoskeletal system susceptible to injury from demanding, repetitive, and prolonged procedures [1,2]. The occurrence of musculoskeletal disorders (MSDs) among dentistry employees is notably more frequent than in the general population, particularly involving the neck, shoulders, and back [1,3]. A study that included 540 dentists and dental students reported lifetime, 12-month, and 7-day MSD prevalence rates of 98.8%, 92%, and 65.6%, respectively [4], while in their review, Hayes et al. [1] found a prevalence of musculoskeletal pain ranging from 64% to 93%. Following with global trends, German prevalence rates of MSDs mirror international findings [5]. These statistics emphasize the need for effective interventions to mitigate MSDs among these professions.

Students in the field of dental health spend extended periods in preclinical laboratories practicing on phantom heads, often adopting incorrect postures for several reasons [2]. First, they must coordinate their position relative to the assistant to ensure a smooth workflow. Second, they must contort themselves to obtain an optimal view of the patient’s teeth inside the mouth. Third, they must position themselves in a manner to ensure patient comfort, which is another crucial factor that requires unconventional postures [6]. Continuously placing unidirectional and asymmetric strain on the joints, which arises due to the lateral sitting position in relation to the patient and the resulting unequal ratio of leverage of the extremities, can lead to muscle imbalances or structural tissue damage, ultimately resulting in lower back pain [2,7]. Therefore, ergonomic work plays a crucial role in the daily activity of dental professionals [2,6].

Current ergonomic training interventions have already been successful for dentists and dental students [3,8]. Lietz et al. [3] summarized three studies evaluating the effects of ergonomic training interventions, all of which showed promise toward the reduction of MSDs in dentists and dental students, although none of the studies included dental hygiene or assistant students [8,9,10]. Further studies have investigated the benefits of magnification loupes [11] and arm supports [12], as well as their effects on working posture [11] and muscle activity [13]. The scaling task by La Delfa et al. [13] evaluated the excessive strain on the neck extensors and stabilizing muscles of the scapula at low levels and concluded that physical demands varied depending on the assistant’s working position relative to the manikin’s head. Lindegård et al. [14], Hayes et al. [15,16], and Smith et al. [17], who worked with dental hygienists, assessed the effects of prismatic glasses and dental instruments on the symptoms of MSDs. Female employees often experience heavier domestic roles and less frequent rest periods and are, therefore, more at risk of developing work-related MSDs [8]. Dental hygienists and assistants, who are typically positioned on the left side of the patient, assume responsibility for suction and holding tasks, which are typically performed while seated. They are exposed to high postural loads because they perform a significant amount of static and holding work [18]. Students have an entire professional life ahead after school, meaning that significant health benefits may be created through professional training early in their education. Moreover, it is expected that institutions will assist in this endeavor by implementing ergonomic practices that benefit their students [2]. Further investigation is required to fully understand the significance of ergonomic interventions in the role of dental assistants.

Numerous observational methods, such as the Rapid Upper Limb Assessment (RULA), Ovako Work Posture Assessment System (OWAS), and Rapid Entire Body Assessment (REBA), enable ergonomists to measure the postural load by providing a risk score for MSDs [19,20]. Kee [21] compared RULA to OWAS and REBA for the assessment of postural loads and concluded that RULA may be the best system for estimating postural load. However, an inherent limitation is that observational assessment procedures require not only the involvement of a field expert for labor-intensive manual analysis but also rely heavily on the subjective judgment of the evaluator, potentially resulting in substantial variability among different raters [22]. As one of the most frequently used observational methods, RULA is applicable for data collection with inertial measurement units (IMUs) [20,23] and has proven to be advantageous for recording postural load throughout the work process [24]. Maurer-Grubinger et al. [18] demonstrated the benefits of using IMUs in the field of dentistry by comparing two different work routines and delivering an approach for the objective and detailed ergonomic analysis of various RULA levels. Ohlendorf et al. [25] used an IMU for motion-capturing dentists and dental assistants and concluded that the working posture may be determined more by working habits than by the arrangement of dental equipment and tools. They also emphasized that field investigations using IMUs are required in the observation of ergonomics.

Building upon these research deficits, the present study evaluated the effects of a specific educational ergonomic lecture using RULA. Additionally, the prevalence of MSDs was recorded, and the influence of a specific stretching and strengthening training intervention over 5 weeks was assessed to determine if it had a positive effect on work posture [26] and work-related MSDs [27].

Therefore, the authors hypothesized that:Educating dental assistant students on ergonomic principles for use in the workplace would lead to a significant improvement in their working posture.A targeted training intervention (stretching and strengthening) lasting 5 weeks, provided by an ergonomic assessment application, would allow for the short- and mid-term reduction in the occurrence of MSDs.

## 2. Materials and Methods

### 2.1. Experimental Design

The present study was a randomized controlled trial (RCT) conducted with a parallel-group and pre-post-test design (Figure 1) and was initiated in January 2024. Each participant was informed verbally and in writing of the protocol for the present study, and signed an informed consent form regarding data rights, recorded videos, and publishing the results of the study procedures. The study was conducted in accordance with the guidelines of the Declaration of Helsinki and approved by our institutional ethics committee (Ethikkommission RPTU Kaiserslautern-Landau, Nr. 66-2023).

### 2.2. Participants

Eighteen dental assistant students (all female) from Worms BBS School (Germany) voluntarily participated in this experimental study (age, 19.44 ± 6.83 years; height, 164.59 ± 5.32 cm; weight, 64.88 ± 16.52 kg; body mass index [BMI], 19.70 ± 4.89 kg/m^2^). The participants were randomly divided into two groups: the intervention group (IG), who underwent a five-week strengthening and stretching training routine four times a week, and the control group (CG), who underwent no training (Table 1). All participants attended the ergonomic lectures and underwent workplace analyses.

Professional dental assistants are characterized as follows [5]:They usually remain in the same position for long periods because of the monotonous work (holding and suctioning) performed.Frequently spend long periods in a chair without a break because of patient preparation and follow-up (e.g., removal of temporaries and impressions).Sit subordinate to the position of the dentist.Frequently encounter a poor field of vision, as the mouth is a small, detailed working area (e.g., for fillings, the dentist must first and foremost be able to see well, and the dental assistant must adapt to the dentist’s position).Require additional equipment to perform their work (e.g., magnifying or prism loupes and armrests on chairs).

The inclusion criterion was defined as follows: full-time dental assistant student with a minimum of three months’ experience in the current professional segment. The exclusion criteria were defined as follows: current injuries to the musculoskeletal system, acute restriction of physical activity, and/or surgical treatment of the musculoskeletal system in the previous four weeks. During the intervention period, three participants who were unable to complete the post-test were excluded.

### 2.3. Procedure

The study was conducted as follows: (1) The participants completed a Nordic Questionnaire to identify the presence of any MSDs; (2) they underwent a IMU-based motion capture (MoCap; Movella, Enschede, The Netherlands) while working on a patient in a dental practice; (3) then, a lecture on ergonomics was provided; (4) the experimental group completed a 5-week training intervention; (5) after the intervention, the participants underwent a follow-up IMU-based MoCap at the same time of the day as in the pre-test; and (6) after 18 weeks, the participants completed follow-up questionnaires regarding MSDs (Figure 2).

After the data collection was finalized, the data were prepared for further analysis (see Section 2.10).

### 2.4. Measurement of Musculoskeletal Disorders (MSDs)

The German-adapted version of the Nordic Questionnaire, originally created by Kuorinka et al. [28], was used to measure musculoskeletal discomfort. In the questionnaire, discomfort was defined as “stinging, pain, or discomfort (tingling or numbness) in the respective body regions”. Based on the results of Liebers et al. [29], the NFB*MSB questionnaire largely meets the expectations of practicability, test-retest, and content validity. The FB*MSB is used to determine where and how often musculoskeletal complaints occur, and whether they restrict activities at work or during leisure time. As the questionnaire has a modular selection option, 4-week and 7-day prevalence rates were used. The period prevalence was determined, representing the quotient of the number of “yes-answers” in relation to the total number of answers for the period under consideration [29,30].

Additionally, visual analog scales (VASs) [31,32] were used to assess discomfort in six different body areas in the pre- and post-tests, as follows:NeckUpper armsLower armsWristsTrunkLower back

### 2.5. Experimental Settings

Webcam (Logitech, Apples, Switzerland) and tablet (iPad; Apple, Cupertino, CA, USA) cameras and an Awinda station (Movella, Henderson, NV, USA) were positioned approximately 2.50 m from the patient to guarantee good video quality and an optimal view (Figure 3). The height and distance of the tablet (Apple iPad, Apple, USA) were standardized (height: 88 cm; distance: 252 cm). All recordings were made by the same two ergonomists, each with several years of experience in the health sciences. Each subject could adapt the seat height and patient position according to their individual preference.

### 2.6. Experimental Task

All dental assistant students performed the same experimental tasks. Both pre- and post-test measurements were carried out at the same time of the day (between 11 a.m. and 2 p.m.). The task was a one-off task for the sample. The test supervisor read each task to the participants before the measurements began as follows:Sit in the assistant chair.The dentist wants to fill tooth 36.Please hold off the cheek with the mouth mirror and the tongue with the big aspirator tip.You may try out the position once.Now, perform the task.

This sequence represents a typical task for a dental assistant, which must be performed frequently throughout their everyday work. Dental assistants generally adopt many static postures in addition to holding tools in each patient’s mouth.

### 2.7. IMU-Based MoCap

The participants underwent a 90 s MoCap using IMUs (Figure 4). Each IMU comprised of a three-axis accelerometer (±16 g), three-axis gyroscope (±2000 degrees/s), and three-axis magnetometer (±1.9 Gauss) [33]. These axes represent a robust and precise reference system for reconstructing three-dimensional (3D) motion in the workplace [34] and can deliver repeatable and accurate ergonomic risk scores [20,35].

The use of IMU to recognize employee exertion has increasingly become a focus in occupational science [20]; nevertheless, missing angle thresholds such as upper arm abduction or neck twisting must be considered, and IMU can experience magnetic disturbances [24]. Algorithms that use IMU data to provide score-based results are an option, and several proposals have recently been published [33].

The “upper body” suit configuration (Xsens MVN Analyze Pro 2024.2; Xsens Technologies B.V. (Enschede, The Netherlands)) was used, which means that eleven inertial measurement units were attached to the following body parts:HeadSternumShoulder (left and right)Upper arm (left and right)Forearm (left and right)Hand (left and right)Pelvis

### 2.8. Rapid Upper Limb Assessment Score

Within the RULA scheme, posture, muscle engagement, and external loads affecting various body regions, such as the neck, trunk, and upper limbs were evaluated using partial scores for each anatomical region (upper arm, lower arm, wrist, neck, trunk, and legs), as follows:(1)Part-score A was determined based on the arms and wrists, muscle activity (repetition or static posture > 1 min), and forces (<2 kg, 2–10 kg, >10 kg; repetitive or static);(2)Part-score B was determined by the neck, trunk, legs, muscle activity (see above), and forces (see above); and(3)The final score (C) was based on part-scores A and B, and reflects the MSD risk level—the final scores ranged from 1 to 7, where a score of 1 and 2 indicated low risk, scores of 3 or 4 indicated a potential necessity for intervention or procedural modifications, scores of 5 or 6 implied an impending need for alterations, and a score of 7 denoted a pressing requirement for a change in work procedures [36]. The use of IMUs makes it possible to map the joint angles over the entire work process [18].

### 2.9. Ergonomics Lecture and Training Intervention

The interventions in the present study included two main aspects that were oriented and slightly adapted from Dehghan et al. [9], as follows:Education on ergonomics—Dental assistant students attended a multifaceted ergonomic lecture covering the fundamental principles of ergonomics, ergonomic risk factors specific to the role of the dental assistant, and components of an ergonomic intervention program, presenting a balanced posture according to Lindegård et al. [37], in which three exercises are performed daily, focusing on the shoulders and neck.Workstation adjustment—During each session, the dental assistant students’ working conditions were assessed directly on the job, which may be the most effective in achieving practical results [8], while ergonomic risk factors were also identified. The participants were guided to adapt their workstations based on the ergonomic risk factors prevalent in the role of dental assistants, which involved providing instructions on proper posture and equipment alignment to ensure optimal working conditions.

The training intervention included four weekly sessions over 5 weeks. During each session, six exercises were completed in three sets of ten repetitions. The training exercises were produced via the Ergofreude health application, and quick response (QR) codes were available to use the video-guided exercises from each participant’s smartphone. The training included stretching exercises for the neck-shoulder region, and the focus of these exercises was on the trunk and neck regions, which, according to scientific studies, are the regions of the body with the highest prevalence of MSDs, as previously discussed. Two health science experts instructed the intervention group (n = 9) on all the exercises prior to the start of the intervention. Additionally, the participants were instructed to document their daily physical activity using the Ergofreude health application.

### 2.10. Data Processing and Analysis

The relevant ergonomic kinematic variables were rated based on the RULA score, which investigates the exposure of workers to risk factors associated with work-related disorders, using a self-written MATLAB script (MathWorks, Natick, MA, USA) [38]. Starting from the joint angles measured by the 11 IMU, angle-to-score mapping was used at each time point during the experimental task (90 s). Working time in (“acceptable”, “measures should be initiated in the near future”, “measures should be initiated shortly”, and “measures should be initiated immediately”) for each posture score could be calculated over the whole working process to assess a final score that represented the whole working process instead of a subjective rating.

Posture, muscle engagement, and external loads affecting distinct body regions, such as the neck, trunk, and upper limbs, were evaluated using partial scores for each anatomical region (upper arm, lower arm, wrist, neck, trunk, and legs) [39]:A: Upper and lower arms and wrists + muscle activity (none = 0; repetition or static posture > 1 min = 1) and force (<2 kg = 0; 2–10 kg temporary = 1, 2–10 kg static or repetitive = 2; >10 kg repetitive or sudden = 3). Muscle activity was set to 0, and force was set to 1 in both measurements.B: Neck, trunk, legs + muscle activity (see A above) and force (see A above). Muscle activity was set to 1, and force was set to 0 in both measurements.

A scoring system was designed to calculate the total score for each frame, allowing for subsequent statistical analyses (Figure 5). The leg value was set to 1.

First, all data were checked for normal distribution using the Shapiro–Wilk test in SPSS v29 (IBM, SPSS Inc., Chicago, IL, USA), and a visual analysis of data was performed to detect outliers. The pre- and post-test RULA scores were not normally distributed. Simulation studies have shown that a mixed analysis of variance (MANOVA) is largely robust to violations of the normal distribution assumption [40] when normality is the only assumption violated [41]. To examine group differences (between-subject-factor), differences between time of measurement (within-subject factor), and the interaction effect of both factors, MANOVA and Bonferroni post hoc analysis were conducted using SPSS (IBM, version 29, SPSS Inc., Chicago IL, USA). Adjusted *p*-values, determined by MANOVA, were compared with an alpha level of 0.05, and the effect size was assessed following Cohen [42]. Furthermore, the Greenhouse–Geisser adjustment was used to correct violations of sphericity. Additionally, six repeated-measures analyses of variance (rmANOVAs) were performed using R. Additionally, *t*-tests for independent samples were used to detect differences in age and sports activity between the control and intervention groups. Levene’s test confirmed the homogeneity of the data variance. Visualization of RULA scores was performed using the Python library “Seaborn” [43].

For MSDs (VAS and Nordic Questionnaire), the Shapiro–Wilk test showed a normal distribution of data, although not for elbows pre-/post- and thoracal-post. An additional MANOVA was performed using SPSS (between-subject factor, group; within-subject factor, time of measurements [pre-test, post-test, follow-up]). The Greenhouse–Geisser correction was used to correct for violations of sphericity.

## 3. Results

### 3.1. Rapid Upper Limb Assessment (RULA) Scores

Mixed ANOVA results showed no significant interaction between groups and time of measurements (Greenhouse–Geisser: F(1,6) = 0.226; *p* = 0.651; η_p_^2^ = 0.036). There was a significant reduction between the time of measurements (F(1,6) = 34.940; *p* = 0.001; η_p_^2^ = 0.853), but no significant difference between the groups (F(1,6) = 1.034; *p* = 0.348) (descriptive data in Table 2 and Figure 6 and Figure 7). Bonferroni-adjusted post hoc analysis revealed a significant (*p* = 0.001) score reduction between the time of measurement from pre- to post-test (*Mean*_Diff_ = −0.560; 95% confidence interval [CI], −0.792 to −0.328) (Figure 6).

Regarding the single score of each body region, the upper arm (*Mean*_Diff_ = −0.873; 95%CI, −1.495 to −0.250; *p* = 0.002), wrist (*Mean*_Diff_ = −0.819; 95%CI, −1.076 to −0.562; *p* < 0.005), and total RULA (*Mean*_Diff_ = −0.964; 95%CI, −1.651 to −0.278; *p* = 0.017) were significantly lower. The results of the statistical analysis and robust rmANOVAs are shown in Table 3.

Figure 8 shows the posture change of subject 16 after the intervention in a pre-post comparison.

Based on the *t*-tests for independent samples, no significant differences between groups (age and sports activity) were detected (age [years], CG = 20.44; IG = 18.56; *p* = 0.572; activity [h/week], CG = 0.78; IG = 0.44; *p* = 0.422).

### 3.2. Detected Musculosceletal Disorders (MSDs)

Mixed ANOVA with Greenhouse–Geisser correction of discomfort rating (VAS) showed no significant interaction between the time of measurement and the groups (F(1,5) = 2.079, *p* = 0.150). Furthermore, there was no significant difference between the time of measurement (F(1,5) = 4.113; *p* = 0.092) and groups (F(1,5) = 1.394; *p* = 0.291). Table 4 shows the descriptive statistics of the MSDs. 

The descriptive data (Table 4) show a non-significant decrease in the VAS scores in the following body parts: cervical spine, shoulder and arms, thoracic spine, and lumbar spine, with a temporary increase in the VAS scores in elbows and lower arms, as well as wrist and hands. Regarding the 4-week and 7-day prevalences of the sample, the intervention group showed in some body regions more decreases than the control group (neck/cervical spine, shoulder/arms in post-test_1_, wrist/hands in post-test_1_, and lumbar spine in post-test_2_), but there was no significant group difference in the statistical analysis.

## 4. Discussion

### 4.1. Main Findings and Contributions

Statistical analysis showed that the total RULA score (pre, 4.87 ± 1.13; post, 3.67 ± 0.90) was significantly decreased after the ergonomics lecture for dental assistant students at school and at the individual workplace. The results of the present study are in line with current research in the field of dentistry, showing that ergonomic and physical training interventions have had positive effects on the dental profession [3,8,9,10]. Yiu et al. [26] measured a decrease of 1.88 points in RULA after a 10-week exercise intervention. Hayes et al. [15] and Smith et al. [17] focused on dental hygiene students and detected a decrease in postural risk after the intervention. In contrast to the methodology used in the present study, they primarily referred to the effects of dental instruments on MSD symptoms. Consequently, the results of the present study support the idea that the chosen intervention, including a multifaceted lecture on the principles and risks of ergonomics, led to a reduction in the postural risk at work in the target group of dental assistants. IMU data represent a level that indicates “an impending need for alterations” and is reduced to the level of “a potential necessity for intervention or procedural modifications”.

It must be noted that significant differences between the pre- and post-tests can particularly be found in the upper arm and wrist regions. The upper arm value was based on the kinematic data of the shoulder (flexion, abduction, and support). The ergonomic training of the test subjects may have led to habitual adaptation regarding the movement of the shoulder. The alignment of the upper extremity noticeably improved and was supported by the statistical analysis. This may have a positive influence on the shoulder-neck region. Furthermore, an improvement in the trunk score would have been expected. However, this was not shown by the statistical analysis. The reason for this could be that the dental assistants cannot be brought into an upright position by the work task despite the training in an ergonomically positive working posture due to the task and the doctor’s assistance with the patient. In contrast, the demands of the work require that the dental assistants move into a postural position, which can lead to complaints in the lumbar spine and shoulder-neck region. An approach based on behavioral prevention activities, such as physical training (stretching and strengthening) beyond the work process, might be useful.

The wrist score results from pronation and rotation. In this body region, the subjects showed a significantly lower postural risk in the post-test. In particular, the guidance of the medical tool at the patient’s mouth was performed more consciously.

Recent studies and the results of the present study suggest that training regarding health-conscious working practices should be given particular attention in the education of dental assistant students, as they are only briefly mentioned in the German study curriculum [44]. Based on the data obtained from the present study, the authors advocate for lectures on the basics of workplace ergonomics to provide information on both behavioral and situational preventive measures, further contributing to the prevention of complaints in the short, medium, and long term.

Regarding the five-week stretching and strengthening intervention, no significant differences in MSDs were found between the groups or times of measurement. In general, five weeks is a relatively short period to determine the immediate effects of an exercise intervention on musculoskeletal complaints. Owing to time restrictions in the curriculum, it was not possible to extend the duration of the training intervention. Furthermore, the choice of exercise and load parameters may not have considered an individual’s current training status. The load parameters were chosen comparably for all test subjects; however, not every test subject had the same fitness level. Therefore, a statistical analysis was conducted regarding the individual activity levels per week, which showed no significant differences between the groups. Despite completing the training documentation, it cannot be ruled out that the test subjects skipped performing any exercise. After 18 weeks in the follow-up MSD questionnaire, the intervention group educated in an exercise program showed less discomfort, especially in the spine (thoracic and lumbar). Therefore, the data may support the assumption that there could be middle- to long-term effects of additional exercise interventions alongside work to adequately compensate for daily postural load. Nevertheless, it must be emphasized that the statistical analysis did not support this conclusion. Furthermore, the number of voluntary exercise sessions conducted by the participants after the intervention period was not documented. Therefore, the follow-up tests must be viewed critically.

### 4.2. Methods

The choice of study design was a controlled, randomized trial to allow a statistically accurate comparison between subjects who only received ergonomic training and those who underwent an additional training intervention. Based on the chosen study design, it was not possible to control which effect was due to which of the two components of the intervention. With a larger sample size, further groups could be formed to identify the most effective measures. The participants were considered very attentive during the sessions. As the dental assistant students were made aware of poor posture in the workplace, not only the final RULA score, but also the individual body scores were considered in the evaluation to determine which parts of the body could be best influenced by ergonomic education.

For the stretching and strengthening interventions, the participants were randomly divided into the intervention and control groups. With a sample size of 18 participants and three dropouts before the post-test, the size of the groups must be considered low. The lower number of subjects, 15 instead of 18, leads to a greater weighting of individual cases in the statistical analysis and consequently has a negative impact on the validity of the study. Only one test subject in the intervention group dropped out during the post-measurement stage. All other test subjects completed all units in accordance with the training documentation, indicating good participation.

The 90 s recording of the work process using IMUs was carried out once for all test subjects. A multiple measurement of the work process could further strengthen the validity of the study and should be included in future studies. However, regarding the results of the kinematic data from the transverse plane, the measurement errors of the IMU must also be considered. Regarding wrist values, it must be emphasized that the body of evidence for the wrist joint remains conflicting [45]. In their review, Poitras et al. [45] showed that the root-mean-squared error varied from 2.2° to 30°.

Regarding the statistical analysis, a mixed ANOVA and robust repeated-measures ANOVAs were performed. Compared with *t*-tests for dependent samples, these methods represent a higher statistical power. Robust repeated-measures ANOVAs allowed for the analysis of individual body parts; however, group differences between the individual body parts were not further controlled.

### 4.3. Strengths and Limitations

The results demonstrate the importance of health-related lectures within the curriculum and underscore their benefits. Dental assistants face a high risk of developing MSDs if they are not guided to adopt proper posture during early professional education. Enhanced posture has the potential to alleviate current musculoskeletal discomfort and mitigate the risk of developing MSDs. However, given the limited follow-up duration of most studies, it is essential to substantiate this conclusion through further research [3]. Recent studies have demonstrated the importance of ergonomic interventions in the professional field of dentistry [8,10,26,46]. In contrast to the traditional paper-to-pencil approach, assessing RULA with an IMU allows the recording of the work process over a longer period, and data are generated in relation to the joint kinematics of both sides of the body. This may be considered a significant enhancement of the RULA. Especially regarding that this measurement system is not bound to laboratories and enables objective measurements with good quality can help practitioners to monitor possible risk factors on a regular basis.

Only one 90 s recording of the experimental task was carried out at each measurement time. The experimental design can be further strengthened by performing it several times and extending the measurement period. The small sample size and relatively short intervention period (5 weeks) are additional limitations of this study. With larger samples and equal sex distribution, the effects of the intervention on MSDs and work attitudes could change. In future studies, it is recommended that the sample be expanded to further strengthen the validity of the results. Only females were included in this study. This is because the profession is predominantly practiced by women, making it difficult to access male test subjects. Another limitation was the absence of a control group that did not receive any intervention. This raises questions regarding the extent to which the observed reduction in perceived pain was attributable to ergonomic and physical training interventions as opposed to the potential time-dependent adjustments made to cope with the physical demands of work.

### 4.4. Relevance to Industry

The authors aimed to raise awareness among dental assistant students regarding the risks associated with improper posture. The findings suggest that lectures on ergonomics and workplace assessments can have a positive impact on dentists’ work posture and, therefore, prevent work-related MSDs. In addition, the present study delivered promising results to further improve observational risk assessments, such as RULA, in a highly promising target group with high prevalence rates.

### 4.5. Future Research

Future studies should include a control group without ergonomic lectures to determine the isolated influence of this intervention. In addition to a female-based study population, male students should be included to detect gender differences. Furthermore, the sample size should be increased. A 5-week training intervention did not have a significant influence on MSDs; however, based on positive trends in MSD data, future studies should integrate longer intervention periods to fully detect the effect.

## 5. Conclusions

The findings of the present study indicate that ergonomic lectures effectively addressed ergonomic issues among dental assistants. As such, institutions should implement occupational health training focused on preventing MSDs, especially teaching ergonomic principles to benefit students in their early professional careers. Generally, the authors recommend implementing ergonomic training in schools and workplaces at an early stage of education to encourage students in recognizing the importance of healthy exercise even before they start their careers. Furthermore, strengthening and stretching exercises that are practiced regularly over longer periods and focus on the shoulder-neck and lower back regions might be beneficial as a counterbalance to the daily work-related postural load in this occupational field.

## Figures and Tables

**Figure 1 healthcare-12-01670-f001:**
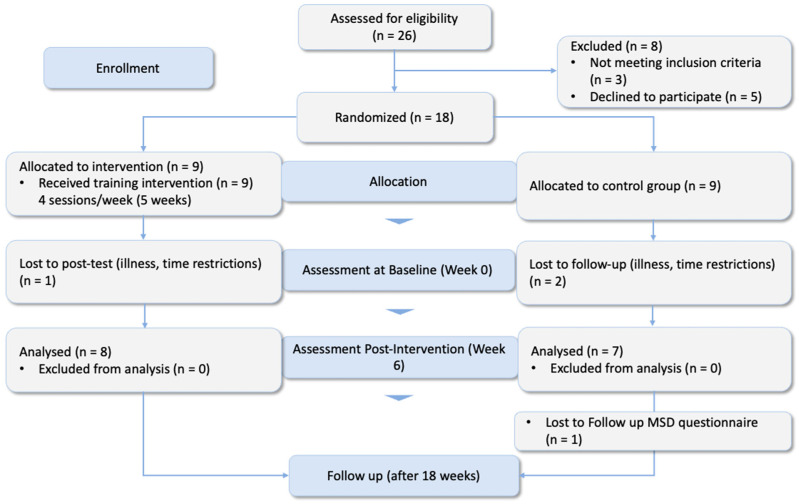
Consort flow diagram of the randomized-controlled trial.

**Figure 2 healthcare-12-01670-f002:**
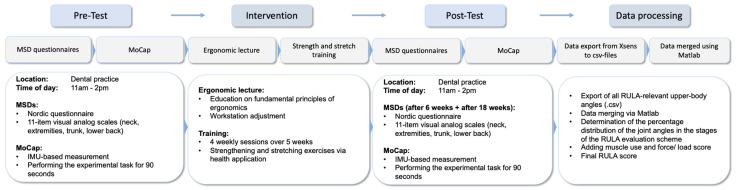
Study procedure.

**Figure 3 healthcare-12-01670-f003:**
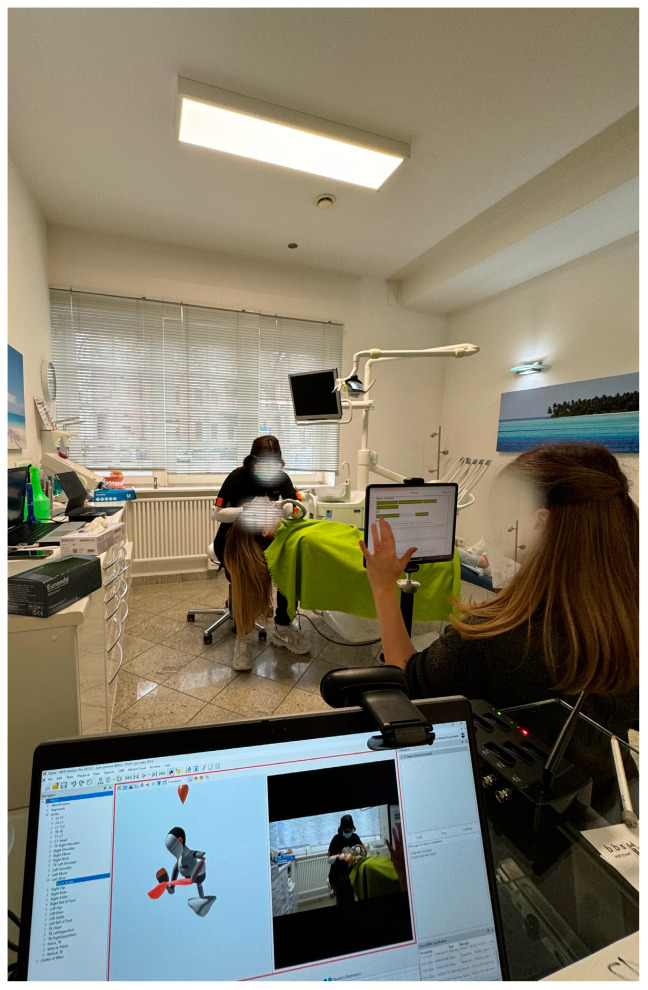
Test equipment (Movella Awinda station and Xsens MVN Analyze Pro 2024.2; Xsens Technologies B.V. (Enschede, The Netherlands)).

**Figure 4 healthcare-12-01670-f004:**
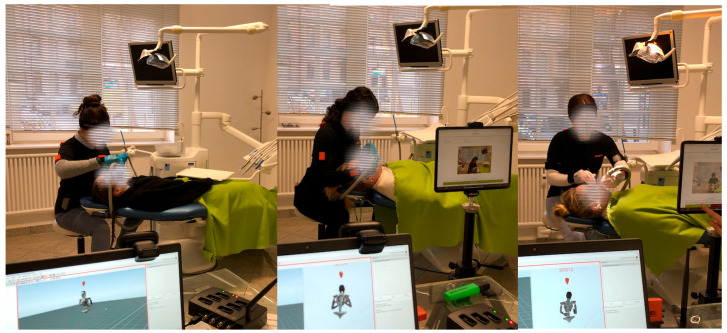
Experimental task (motion capture of dental assistant students working on a patient).

**Figure 5 healthcare-12-01670-f005:**
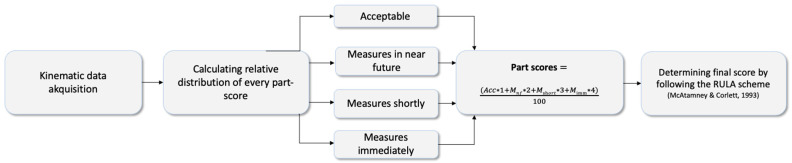
Workflow for determining the final RULA score across the experimental task [39]. Abbreviations: *Acc* = acceptable; *M_nf_* = measures in the near future; *M_shortly_* = measures shortly; *M*_imm_ = measures immediately.

**Figure 6 healthcare-12-01670-f006:**
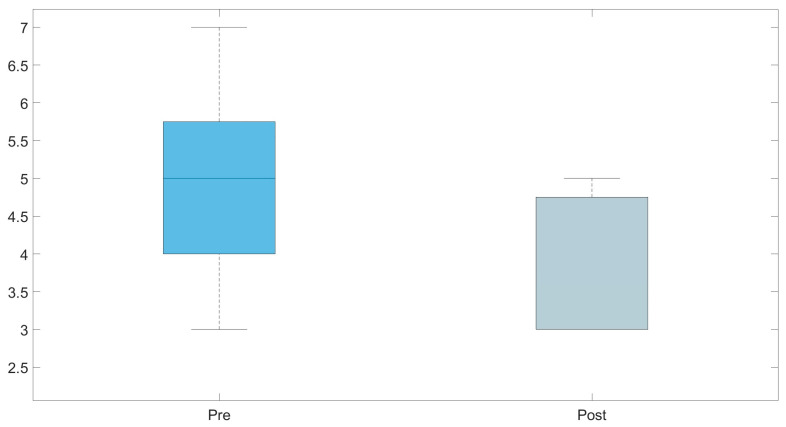
Boxplots representing a comparison of the mean Rapid Upper Limb Assessment (RULA) scores from the pre- and post-test assessments using the IMU-based MoCap.

**Figure 7 healthcare-12-01670-f007:**
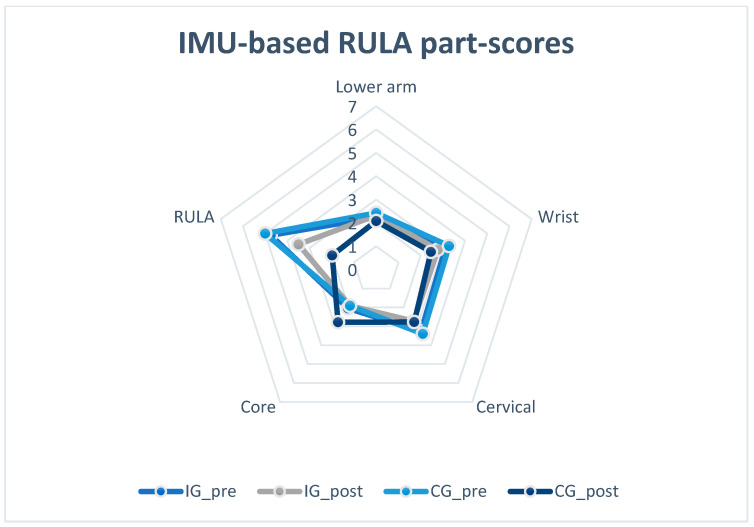
Visualization of RULA part-score results in pre- and post-test. Abbreviations: IG_pre = intervention group in pre-test; IG_post = intervention group in post-test; CG_pre = control group in pre-test; CG_post = control group in post-test.

**Figure 8 healthcare-12-01670-f008:**
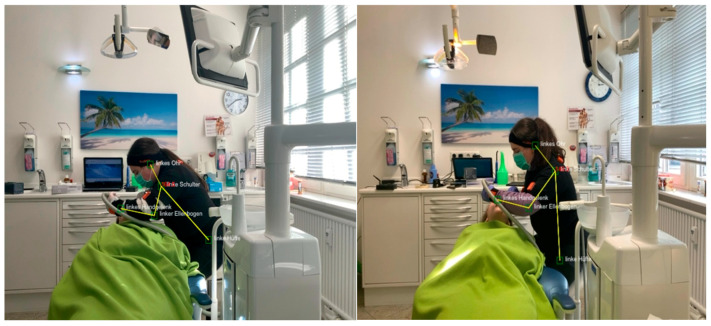
Working postures of subject number 16 (sagittal plane; left: pre-test; right: post-test) (angles shown are pose estimation-based and were not part of the data processing in this study).

**Table 1 healthcare-12-01670-t001:** Anthropometric data from participants randomly divided into intervention and control groups.

	Intervention Group (IG) (n = 9)		Control Group (CG) (n = 9)	
	Mean	SD	Mean	SD
Age (years)	18.67	5.43	20.33	8.22
Height (cm)	165.18	6.81	164.01	3.60
Weight (kg)	64.37	14.09	65.40	19.51
BMI (kg/m^2^) *	19.52	4.48	19.88	5.53

* BMI, body mass index, defined as weight per kilogram divided by the squared body height in meters.

**Table 2 healthcare-12-01670-t002:** Descriptive data of RULA from the intervention group (IG) and control group (CG). Abbreviations: SD = standard deviation.

	RULA_pre				RULA_post			
	Mean	SD	Min	Max	Mean	SD	Min	Max
IG	4.75	0.82	3.00	6.00	3.50	0.96	3.00	5.00
CG	5.00	1.29	3.00	7.00	3.86	0.90	3.00	5.00

**Table 3 healthcare-12-01670-t003:** Results of RULA detected using inertial measurement unit (IMU)-based motion capture (MoCap) (mean, standard deviation, and interaction effects of time of measurements [pre, post] and body region [upper arm, lower arm, wrist, neck, trunk, RULA] in robust rmANOVA). Score interpretation: 1 and 2 = acceptable; 3 and 4 = measures in the near future; 5 and 6 = measures shortly; 7 = measures immediately.

		IMU-Based MoCap RULA Score	Robust rmANOVAs (ToM × Body Region)
Upper arm	Pre	3.00 ± 0.53	*p* = 0.002
	Post	1.98 ± 0.41	
Lower arm	Pre	2.33 ± 0.39	*p* = 0.504
	Post	2.39 ± 0.20	
Wrist	Pre	3.20 ± 0.48	*p* = 0.005
	Post	2.77 ± 0.25	
Neck	Pre	3.33 ± 1.14	*p* = 0.196
	Post	2.75 ± 0.90	
Trunk	Pre	1.99 ± 0.69	*p* = 0.394
	Post	1.95 ± 0.40	
Total RULA	Pre	4.87 ± 1.13	*p* = 0.017
	Post	3.67 ± 0.90	

RULA, Rapid Upper Limb Assessment; IMU, inertial measurement unit; rmANOVA, repeated-measures analysis of variance; ToM, time of measurement; MoCap, motion capture.

**Table 4 healthcare-12-01670-t004:** Descriptive data of MSDs from the intervention and control groups. VAS interpretation [32]: 0–0.4 cm = no pain; 0.5–4.4 cm = mild pain; 4.5–7.4 cm = moderate pain; 7.5–10 cm = severe pain. Nordic questionnaire: 4-week prevalence = with complaints on 8–30 days in the last 12 months (response to question A = “Yes, on 8–30 days” or 2); 7-day prevalence = with complaints on 1–7 days in the last 12 months (response to question A = “Yes, on 1–7 days” or 1).

Body Region		Discomfort Rating		Nordic Questionnaire			
Post_1_ = 5 Weeks Post_2_ = 18 Weeks (Follow Up)		Visual Analog Scale		4-Week Prevalence (%)		7-Day Prevalence (%)	
		IG	CG	IG	CG	IG	CG
Neck/Cervical spine	Pre	4.43 ± 3.95	6.14 ± 3.53	75.00	71.43	62.50	71.43
	Post_1_	4.29 ± 2.93	5.14 ± 3.19	50.00	71.43	50.00	57.14
	Post_2_	2.71 ± 3.50	5.57 ± 3.74	42.90	57.10	28.60	57.10
Shoulder/Arms	Pre	4.00 ± 3.65	3.00 ± 3.65	62.50	57.10	62.50	57.10
	Post_1_	2.72 ± 2.43	2.43 ± 2.82	25.00	71.40	25.00	57.10
	Post_2_	1.71 ± 3.30	4.14 ± 3.72	28.60	28.60	28.60	14.30
Elbows/Lower arms	Pre	0.71 ± 1.89	0.00 ± 0.00	0.00	28.60	0.00	28.60
	Post_1_	2.43 ± 3.25	1.00 ± 2.65	12.50	42.90	12.50	42.90
	Post_2_	0.43 ± 0.79	1.57 ± 2.23	0.00	14.30	0.00	14.30
Wrist/Hands	Pre	1.86 ± 2.91	2.57 ± 3.36	37.50	42.90	25.00	42.90
	Post_1_	2.86 ± 2.54	2.29 ± 2.98	25.00	42.90	12.50	42.90
	Post_2_	1.29 ± 1.98	2.57 ± 1.13	42.90	42.90	14.30	42.90
Thoracal spine	Pre	6.57 ± 3.55	6.00 ± 2.71	50.00	100.00	62.50	100.00
	Post_1_	4.00 ± 2.94	2.00 ± 3.00	37.50	57.10	12.50	42.90
	Post_2_	0.29 ± 0.49	4.71 ± 2.75	14.30	57.10	14.30	57.10
Lumbar spine	Pre	7.00 ± 2.77	6.57 ± 2.57	87.50	100.00	75.00	100.00
	Post_1_	6.57 ± 2.99	5.00 ± 3.32	87.50	100.00	62.50	71.40
	Post_2_	3.00 ± 2.31	6.71 ± 3.86	42.90	85.70	14.30	71.40

MSD, musculoskeletal disorder; IG, intervention group; CG control group.

## Data Availability

Data are contained within the article.

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
