# Peer review of "Measuring the Effect of an Ergonomic Lecture on the Rapid Upper Limb Assessment Scores of Dental Assistant Students Using Inertial Sensor-Based Motion Capture—A Randomized Controlled Study"

_healthcare, 2024, doi:10.3390/healthcare12161670_

Round 1

Reviewer 1 Report

Comments and Suggestions for Authors

Thank you very much for the possibility to review this manuscript. I appreciate your work; however, I must address some suggestions for improving its quality.

First of all, please double-check the punctuation, spelling, and style. I found several grammatical errors.

Please also complete the information and include answers to the questions. Were the measurements taken at the same time of day? 

Was the measurement during the procedure a one-off for the sample? Were multiple 90 s measurements taken?

Please explain the abbreviations used the first time you use them.

Please also consider presenting more figures instead of photos or adding more tables. However, this is only a suggestion on my part.

Comments on the Quality of English Language

Please double-check the punctuation, spelling, and style. I found several grammatical errors.

Author Response

Dear Reviewer, we very much appreciate that you took your valuable time to review this manuscript!

Please find the detailed responses in the document attached and the corresponding revisions in track changes in the re-submitted file that we uploaded.

Reviewer 2 Report

Comments and Suggestions for Authors

The article "Measuring the Effect of an Ergonomic Lecture to Dental Assistant Students’ RULA with IMU-based Motion Capture - A Randomized Controlled Study"; provides relevant information in the scientific field, for which I have the following suggestions:

1. Introduction:

 - Page 2, Line 36-41: The introduction mentions several statistics and previous studies. It would be beneficial to include a summary paragraph connecting these studies to the specific purpose of the article.

 - Page 3, Line 48-50: The phrase "Continuously putting unidirectional strain on the joints can lead to muscle imbalances or structural tissue damage" could be clearer by explaining how this phenomenon occurs specifically in the context of dental assistants.

2. Materials and Methods:

 - Page 4, Line 104-112: The description of the experimental design could benefit from a more detailed flow chart or a numbered list of the study steps.

 - Page 5, Line 121-125: In Table 1, make sure all technical terms (such as BMI) are defined in a table footer for clarity.

 - Page 6, Line 137-139: It would be useful to mention informed consent in the participants section to emphasize the ethics of the study.

3. Results:

 - Page 10, Line 301-303: Table 2 shows the RULA results detected by IMU. It would be beneficial to add a brief explanation below the table that interprets these results in simple terms.

 - Page 11, Line 311-315: In Table 3, descriptive data of MSDs should be accompanied by a clear interpretation of how these results affect the conclusion of the study.

4. Discussion:

 - Page 12, Line 320-324: The discussion of the main findings could be more effective if it included a direct comparison with previous studies and explained in greater detail how this study contributes new ideas or evidence.

 - Page 13, Line 385-390: The small sample limitation is briefly mentioned. It would be helpful to discuss how this might have influenced the results and what steps can be taken in future studies to mitigate this limitation.

5. Conclusions:

 - Page 14, Line 441-443: The conclusion mentions the implementation of occupational health programs. It would be beneficial to specify practical recommendations for educators and administrators in terms of policies and programs.

Author Response

Dear Reviewer, we very much appreciate that you took your valuable time to review this manuscript. All the points you mentioned were clear and helped us to substantially improve the manuscript and the presentation of the research project.

Please find the detailed responses in the document attached and the corresponding revisions in track changes in the re-submitted file.

Reviewer 3 Report

Comments and Suggestions for Authors

The manuscript presents an interesting investigation into the effects of ergonomics intervention on dental assistant students' posture and perceived discomfort. Overall, sound methods are used but several writing issues greatly lower the comprehension of research findings. Specifically, the originality of this study is relatively low and the methods section lacks a significant amount of information.

1. Originality: It is well known that ergonomic training intervention has a positive effect on correcting posture and improving MSD (as the authors mentioned in the discussion). Therefore, the results of this study are somewhat predictable and make it difficult to have originality.

2. The section "Procedure" should be included so that readers can track this experiment's actual procedure.

3. How long one experimental trial lasted is unclear. Even though this was revealed in the discussion section (90 s), this must be explained in the methods section.

4. It is unclear where the IMU sensors were attached to the participants' bodies: The anatomical landmarks where the sensors are attached must be explained to replicate and improve the reliability of this study.

4-1. For the measurement from the IMU sensor (change of angle, speed, acceleration), it is also unclear which and how the measurements are calculated into the kinematic data.

5. l. 203 This description seems to correspond to RULA, but the subject is unclear.

6. The title of Section 2.3 should not be just MSDs, but any title such as Measurement of musculoskeletal discomfort, etc. seems more appropriate. Likewise, the title of section 3.2 and "for MSDs..." in l. 276 are also inappropriate.

7. The title and content of Section 2.6 are unclear. The authors would be able to divide the IMU Mocap part and the RULA part.

8. There is a lack of explanation for the experimental design (i.e., factor design), which makes it difficult to understand why MANOVA and rmANOVA were needed in the data analysis in section 2.9. Since this study tests the effectiveness of ergonomic training intervention, one might think that the t-test is just enough. Therefore, an additional section is needed to introduce the factors tested in this study.

9. The title of Section 3.1 should be Rula score

10. section 4.2 (The title also must be changed) is more of a discussion on the application of IMU for field studies rather than a discussion of the obtained data from the experiments; The authors might exclude the part related to the results from the relevant content and include it in the introduction or methods section.

Comments on the Quality of English Language

Extensive editing of English language required

Author Response

(The authors gave the same response as above.)

Round 2

Reviewer 3 Report

Comments and Suggestions for Authors

The contents are improved based on the reviewer's comments and suggestions.